# CONVERTING DIFFUSIONS TO FLOWS ACCELERATES SAMPLING AND SUGGESTS OVER-CONDITIONING OF CO-FOLDING MODELS ON SEQUENCE

**Nele P. Quast, Fergus Imrie & Matthew I.J. Raybould, Yee Whye Teh, Charlotte M. Deane**
Department of Statistics
University of Oxford
Oxford, UK
`{quast,imrie,raybould,y.w.teh,deane}@stats.ox.ac.uk`

**Niklas Abraham & Aaron Schöne**
SimTech
University of Stuttgart
Stuttgart, Germany
`niklas.abraham@ibc.uni-stuttgart.de, aaron.schoene@uni-stuttgart.de`

## ABSTRACT

Deep generative models can predict protein structures from sequence with high accuracy; however, sampling from these models remains computationally burdensome, with current protocols using hundreds of iterations through the trained model to obtain a final predicted structure. To accelerate sampling and improve the interpretability of the prediction trajectories, we convert the stochastic diffusion sampling process into a deterministic flow process. We show that the conversion of pre-trained, diffusion-based structure prediction models to probability-flow ODEs yields equivalent performance on the FoldBench benchmark alongside a 20x sampling speed-up. Furthermore, we demonstrate the effects on prediction diversity and use the intermediate predictions made along the de-noising trajectory to show that deep generative structure prediction methods are strongly conditioned on the sequence and MSA embeddings, appearing to make predictions with weak sensitivity to the noise initialisation. Finally, we discuss the implications of strong sequence conditioning for generative protein structure prediction and protein design, as well as pointing to future experiments that build on our initial results.

## 1 INTRODUCTION

The latest generation of protein structure predictors use diffusion models (Abramson et al., 2024; Passaro et al., 2025; Discovery et al., 2024; The OpenFold3 Team, 2025) to iteratively predict atomic coordinates from noise, conditioned on the amino acid sequence of a protein and its multiple sequence alignment (MSA). These generative methods also enable conditioning on non-amino-acid molecules, such as DNA and drug-like ligands, and are the current state-of-the-art for protein-ligand co-folding (Abramson et al., 2024). During inference, diffusion models integrate the reverse stochastic differential equation (SDE) using the parametrised and trained score function. This requires iterative sampling of the neural network which makes inference costly: current methods take 200 diffusion steps by default to produce a prediction. Furthermore, the integration of SDEs introduces stochasticity into the inference process by design, which can obfuscate the interpretation of the sampling trajectories.

Flow-based models sampled with deterministic ordinary differential equation (ODE) integrators provide an alternative generative framework (Chen et al., 2019). Furthermore, it has been shown that the marginal distributions of the sampling trajectories of diffusion- and flow-based methods are equivalent under certain conditions and can be sampled using the same score function (Karras et al., 2022; Song et al., 2021). Converting the stochastic diffusion SDE into a deterministic flow-based ODE

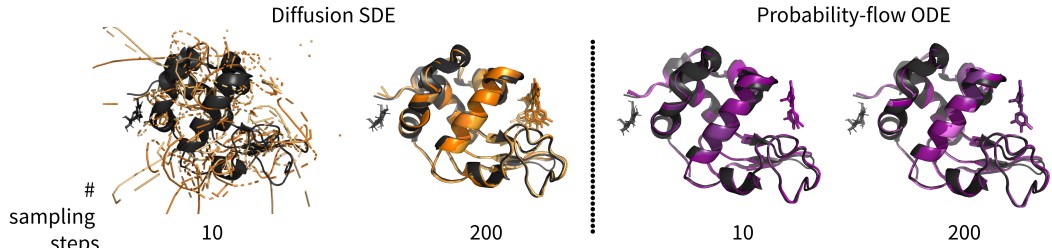

Figure 1: Structure predictions made using diffusion-SDE or PF-ODE sampling with the pre-trained Boltz-2 (Passaro et al., 2025) score function, with the sampling trajectory discretised into 10 or 200 steps.

offers a compelling strategy to accelerate sampling and improve the interpretability of prediction trajectories. In this work, we show that by converting diffusion-based protein structure prediction models into probability-flow ODEs (PF-ODEs) we can accelerate sampling without compromising prediction accuracy. In addition, our analysis of the flow trajectories and intermediate predictions reveals that generative co-folding models are strongly conditioned on sequence information.

Our contributions are threefold:

- We leverage the established theoretical equivalence of the marginal densities of diffusion and probability flow-based models to derive a flow ODE for protein structure prediction and co-folding.

- We provide a deterministic protein structure and complex sampling algorithm that uses the pre-trained score models of the open-source methods Boltz-2 (Passaro et al., 2025) and OpenFold3 (The OpenFold3 Team, 2025) and demonstrate equivalent accuracy with a 20-fold speed-up of inference.

- We demonstrate that generative protein structure prediction methods are strongly conditioned on the input embeddings, and discuss the implications of this for structure prediction and protein design.

## 2 BACKGROUND

### 2.1 OPEN-SOURCE DIFFUSION-BASED STRUCTURE PREDICTION MODELS

Open-source protein structure prediction models such as OpenFold3 (The OpenFold3 Team, 2025) and Boltz-2 (Passaro et al., 2025) are independently retrained implementations of AlphaFold3 (Abramson et al., 2024). Under this framework, the atom coordinates of amino acids, ligands, and nucleotides, denoted $x \in \mathbb{R}^{3 \times N}$, are initialised from Gaussian noise and iteratively updated with the score model conditioned on the protein sequence, its MSA, and ligand embeddings. These diffusion models follow the EDM framework, whereby the atom coordinates $x$, the score, $\nabla_x \log p(x)$, the score model, $s_\theta$, and the denoising model $D_\theta$, are related as follows (Karras et al., 2022):

$$\nabla_x \log(p_t(x)) \approx s_\theta(x, \sigma_t) = \frac{D_\theta(x, \sigma_t) - x}{\sigma_t^2}, \tag{1}$$

, where $p_t(x)$ is defined as $p_t(x) = \int p(x|x_0, \sigma_t) p_{data}(x_0) dx_0$, with a Gaussian noising process $p(x|x_0, \sigma_t) = \mathcal{N}(x|x_0, \sigma_t^2 I)$. Models are trained by predicting the de-noised structure, $\hat{x}_t$, from the noised coordinates and regressing using an $L_2$ loss term. Additional loss terms related to atom bonds and LDDT scores (Mariani et al., 2013), as well as multiple training stages, further improve model performance (Abramson et al. (2024), AlphaFold3 SI Eq. 6). This paradigm means that a prediction of the final de-noised structure, $\hat{x}_t$, is made at every reverse diffusion step. During inference, structure coordinate trajectories are sampled from noise using a discrete stochastic update function (see Appendix A for details). Expanding the terms of the update equation recovers the original formulation of the reverse diffusion SDE under the variance exploding (VE) regime (Song

et al., 2021), relating the approximate score to the coordinate updates[1].

$$dx = -\sigma_t \dot{\sigma}_t s_\theta(x, \sigma_t) dt + \sqrt{\sigma_t \dot{\sigma}_t} dw(t) \tag{2}$$

## 2.2 CONDITIONS FOR EQUIVALENT MARGINAL DENSITIES OF DIFFUSION AND FLOW MODELS

When trained to approximate the same data, the final marginal distribution, $p_0$, sampled using diffusion or flows is identical. Trivially, the initial noise distribution, $p_1$, which is usually chosen to be Gaussian white noise, is also identical. Score-based diffusion models produce samples by taking decreasingly noisy steps along the gradient of the log-likelihood of the data distribution (Song & Ermon, 2019). This reverse SDE is governed by the drift term $f(x, t)$ and the diffusion term $g(t)$.

$$dx = \big(f(x, t) - g(t)^2 \nabla_{x_t} \log p_t(x)\big) dt + g(t) dw(t) \tag{3}$$

Similarly, flow-based models sample the data distribution by evaluating the flow along a deterministic trajectory from the initial noise sample (Chen et al., 2019). In order to guarantee identical marginal distributions $p_t(x)$ for all $t$ between the trajectories defined by the reverse SDE and the probability-flow ODE, the ODE must satisfy the continuity equation, resulting in the reverse ODE process described by Equation 4 (Song et al., 2021). The full derivation via the Fokker-Planck and continuity equations is well outlined by Lai et al. (2025).

$$\dot{x} = f(x, t) - \frac{1}{2} g(t)^2 \nabla_x \log p_t(x) \tag{4}$$

## 2.3 CONDITIONING ON SEQUENCES

Conditioning on an input variable, $c$, enables generative models to represent and sample from the conditional distribution $p(x|c)$, rather than the marginal data distribution $p(x)$. This allows the generative process to be controlled via an input 'prompt'. While multiple methods for conditioning have been proposed (Zhan et al., 2025), diffusion-based structure prediction models learn the conditional score $\nabla_x \log p(x|c)$ directly during training by including embeddings of the protein sequence, its MSA, and ligand identities as inputs to the score model (Abramson et al., 2024).

## 3 DETERMINISTIC SAMPLING FROM PRE-TRAINED PROTEIN STRUCTURE PREDICTION MODELS

### 3.1 PROBABILITY-FLOW IMPLEMENTATION OF PRE-TRAINED PROTEIN STRUCTURE PREDICTION MODELS

After including the conditioning variable, it follows from Equations 1, 2, 3, and 4, that the probability-flow ODE (PF-ODE) describing the same marginal distributions as diffusion-based structure predictors is given by Equation 5 (see Appendix B for the complete derivation).

$$\dot{x} = -\frac{1}{2} \frac{\dot{\sigma}_t}{\sigma_t} (D_\theta(x, \sigma_t|c) - x), \tag{5}$$

with reverse drift and diffusion terms $f(x, t) = 0, g(t) = \sqrt{\sigma_t \dot{\sigma}_t}$. To create a flow-based protein structure prediction model, we implemented an alternative deterministic sampling scheme for the PF-ODE (Equation 5) using the pre-trained score function in Boltz-2 (Passaro et al., 2025) and OpenFold3 (The OpenFold3 Team, 2025). The sampling algorithm (Appendix C Algorithm 1) is as close as possible to the original diffusion sampling implementation (AlphaFold3 SI Algorithm 18 (Abramson et al., 2024)) to maintain comparability.

---

[1]AlphaFold3, Boltz-2 and Openfold3 implementations use additional hyperparameter scaling terms which confound Equation 2 somewhat and lead to a discrepancy in the scaling of the Wiener process, but broadly this equation governs sampling of diffusion-style protein structure prediction models. We found the scaling terms did not effect the flow implementation empirically. Please see Appendix A for details.

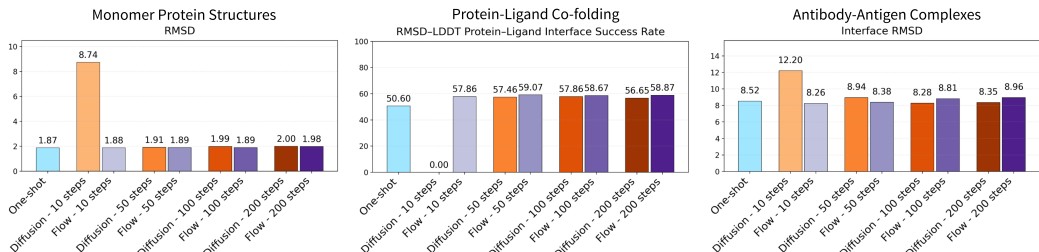

Figure 2: FoldBench Benchmark results comparing PF-ODE (flow) to diffusion-based sampling using pre-trained Boltz-2 score model. Extended results in Appendix F.

## 3.2 Inference with diffusion- and flow-based models

We compared the performance of the diffusion-SDE model to the PF-ODE model conditioned on the sequence, MSA and ligand embeddings, but without additional guidance terms such as FK-steering (Passaro et al., 2025). To investigate the effects of step size, we discretised the noise schedule into 10, 50, 100, and 200 steps. The original diffusion protocol uses 200 steps. We also investigated sequence-conditioning by extracting 'one-shot' predictions: the de-noised predictions $\hat{x}_t$ made during inference[2]. We evaluated the intermediate predictions made on the first step of a ten-step flow trajectory, which receives as input the completely noised coordinates, $x \sim \mathcal{N}(0, \sqrt{\sigma_{\text{data}}}I_3)$, $\sigma_t$, and the sequence-, MSA- and ligand-derived embeddings used as the conditioning variable, $c$.

We used the FoldBench dataset (Xu et al., 2025) to compare our deterministic probability-flow-based structure prediction method to diffusion-based sampling. This dataset includes 330 examples of monomeric proteins, 558 protein-ligand complexes, and 279 multimeric protein complexes, of which 172 are antibody:antigen complexes. We evaluated prediction accuracy over a variety of common metrics, including the root-mean-squared-distance (RMSD), interface RMSD, template-modelling (Tm) score, and the 'RMSD-LDDT protein-ligand interface success rate', as defined in the original benchmark (Xu et al., 2025). As noted by the original authors of FoldBench, Boltz-2 has a later training date cut-off than the FoldBench test-set (Xu et al., 2025). As we are comparing the sampling process using the pre-trained score model this does not effect our comparison of diffusion- to flow-based sampling. We have characterised how many test-set data points are effected in Appendix G.

## 4 Results

### 4.1 Deterministic flow-based sampling maintains accuracy while reducing inference time

The performance on the FoldBench benchmark demonstrates that our flow-based implementation of Boltz-2 performs indistinguishably from the original diffusion model, using a fraction of the inference steps (Fig. 2). As expected, the compute time required to generate predictions scales linearly with the number of sampling steps (Appendix D Fig. 4), yielding a 20x speed-up using flow-based sampling without compromising prediction accuracy. Our flow-model achieves parity in just ten integration steps, while the diffusion model produces noisy predictions at this discretisation (Fig. 1, Appendix E Fig. 5 & 6). Beyond 50 integration steps both diffusion- and flow-based sampling perform similarly, with no further gains observed from further discretising the sampling schedule up to the default 200 steps. Extended results can be found in Appendix F.

### 4.2 Diversity is limited regardless of sampling method

Evaluating the PF-ODE is deterministic apart from the initial condition set by the starting coordinate positions. This could plausibly reduce the sampling diversity of final predictions compared to stochastic sampling, where noise is injected throughout the trajectory. However, empirically, we find

---

[2]We take $\hat{x}_t$ to mean the prediction of the de-noised coordinates $x \sim p_0$, made at time $t$ in the sampling trajectory.

| | Flow | | | | | | | | Diffusion | | | | | | | |
|---|---|---|---|---|---|---|---|---|---|---|---|---|---|---|---|---|
| # integration steps | 10 | | 50 | | 100 | | 200 | | 10 | | 50 | | 100 | | 200 | |
| Cluster threshold (Å) | 1 | 5 | 1 | 5 | 1 | 5 | 1 | 5 | 1 | 5 | 1 | 5 | 1 | 5 | 1 | 5 |
| Mean average | 1.0 | 1.0 | 1.0 | 1.0 | 1.0 | 1.0 | 1.0 | 1.0 | NA | NA | 2.3 | 1.0 | 2.3 | 1.0 | 2.3 | 1.0 |
| Mode average | 1 | 1 | 1 | 1 | 1 | 1 | 1 | 1 | NA | NA | 1 | 1 | 1 | 1 | 1 | 1 |

Table 1: Number of non-redundant ligand poses explored across 25 independent samples, when using different clustering (i.e. redundancy) thresholds.

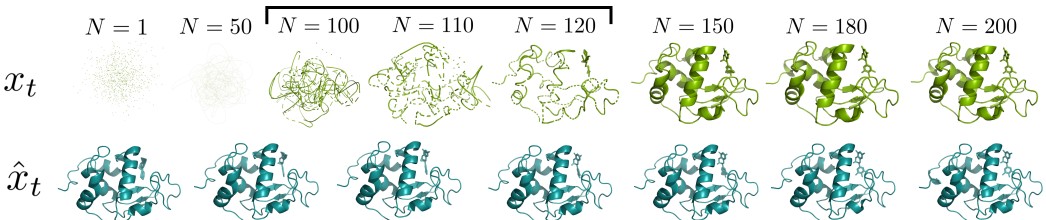

$$x_t$$

$$\hat{x}_t$$

Figure 3: Flow trajectory, $x_t$ and intermediate structure predictions throughout trajectory $\hat{x}_t$.

that the diversity of structure predictions is low regardless of the sampling method (Table 1). We investigated this by examining the number of distinct binding pockets of ligands predicted across 25 samples using both the flow- and diffusion-based sampling schemes of Boltz-2. In summary, we find that neither method samples multiple binding pockets using the default hyperparameters, but that stochastic sampling modestly improves the exploration of local conformations of the ligand within a pocket. Future work should expand the test set and explore the effect of the temperature parameter on sample diversity.

### 4.3 Score-based protein structure prediction models are over-conditioned on sequence inputs

Diffusion-based structure predictors make intermediate, 'one-shot', predictions of the de-noised structure, $\hat{x}_t$ at every reverse diffusion step. The coordinate trajectory using flow-based sampling reveals a crucial structure formation phase approximately halfway through the sampling trajectory, where the noisy coordinates form the final structure (Fig. 3). In the final inference phase the protein structure does not change, but the atoms of the ligand are refined to form a physically-plausible molecule. Examining the intermediate predictions $\hat{x}_t$ reveals that the protein structure predictions and global ligand position do not change throughout the trajectory, regardless of the noise-level. Furthermore, these 'one-shot' predictions perform almost as well as the final sampled predictions on the FoldBench benchmark (Fig. 2). Together, this implies that generative protein structure prediction models rely primarily on the conditioning input, rather than the noisy coordinate position $x_t$ or the noise level $\sigma_t$.

## 5 Conclusion & Discussion

Our results demonstrate that flow-based sampling performs identically to diffusion-based sampling on the FoldBench benchmark, and that by using the PF-ODE formulation the number of sampling steps can be reduced 20-fold from the default settings without effecting prediction accuracy. The diversity of predictions is slightly reduced, however, our initial investigation of protein-ligand complex predictions indicates that neither diffusion- nor flow-based sampling lead to multiple binding pockets being explored. Furthermore, our benchmark of 'one-shot' predictions indicates that the de-noising model is able to make accurate predictions early in the trajectory when the atomic coordinates are dominated by noise. This implies that the model primarily attends to the sequence and ligand input embeddings, rather than the noised coordinates.

To a degree, over-conditioning on the sequence inputs is a desirable characteristic of co-folding models; after all, the majority of natural protein sequences fold into a single structure which is fully defined by the sequence. Unlike for images generated from a text prompt, the conditional distribution $p(x|c)$ in these cases is similar to a point-mass distribution, whereby the coordinates $x$

are fully defined by the sequence 'prompt', $c$. This is exacerbated by the training data, which consists primarily of crystal structure data which fails to capture multiple conformations. It is plausible that the de-noising model learns to disregard the noised coordinates and attends primarily to the input embeddings when making predictions. While this may be desirable when predicting protein structures from sequences, co-folding models seek to predict complexes whose binding interfaces can be diverse. Strong conditioning on the input features risks lowering the sample diversity and over-fitting predictions to the training data, potentially precluding the exploration of valid poses. Furthermore, lack of dependence on the noised coordinates and point-mass conditional distributions prevent the exploration of noise-optimisation and -coupling methods using the PF-ODE we have defined (Lee et al., 2023).

Finally, the evaluation of 'one-shot' predictions calls into question the benefit provided by generative co-folding methods compared to regression models such as AlphaFold-Multimer (Evans et al., 2022). Nonetheless, the flexibility of handling arbitrary atomic inputs (such as ligands and DNA), and the ability to incorporate guidance terms during the sampling trajectory make generative models more adaptable and better suited to the incorporation of additional data. Future work should build on our initial findings by exploring whether the diversity of our flow model can be improved by directed noise-initialisation or guidance terms, as well as investigating more sophisticated ODE and SDE integrators to yield further gains in sampling efficiency.

## 6 ACKNOWLEDGEMENTS AND FUNDING

The authors would like to acknowledge the Boltz team and Merck KGaA for organising the hackathon during which some of the co-authors met and began discussions. NPQ is supported by the Engineering and Physical Sciences Research Council [EPSRC, grant EP/S024093/1] and Immunocore Ltd via SABS:R3. FI and NPQ are supported by the OpenBind project, which received funding from the UK Department of Science, Innovation and Technology (DSIT) under grant number G2-SCH-2025-06-16537. MIJR is supported by the Biotechnology and Biological Sciences Research Council (BBSRC) and the Coalition for Epidemic Preparedness Innovations (CEPI). The authors acknowledge the use of resources provided by the Isambard-AI National AI Research Resource (AIRR). Isambard-AI is operated by the University of Bristol and is funded by the UK Government's Department for Science, Innovation and Technology (DSIT) via UK Research and Innovation, and the Science and Technology Facilities Council [ST/AIRR/I-A-I/1023].

## 7 MEANINGFULNESS STATEMENT

In this novel application of established theory to co-folding models, we convert the diffusion-SDE of AlphaFold3-style methods into a probability-flow ODE, and show preserved accuracy while reducing the number of sampling steps from 200 to 10. The presented analysis and benchmark of one-shot predictions suggest that co-folding models are strongly conditioned on the sequence and ligand embeddings, and appear to disregard the noised coordinates; we discuss the implications thereof for sampling and guidance. As co-folding models are integral to the representation of the molecular foundations of life, our research aligns with the objectives of LMRL.

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

## A   EXPANDED SDE IMPLEMENTATION EQUATIONS

In AlphaFold3 the sampling equation is defined in Algorithm 18 of the SI (Abramson et al., 2024). We adapt the notation slightly for legibility and consistency with the notation used by Karras et al. (2022). The sampling schedule is a linear function of the integration step index $t$ given by:

$$\sigma_t = \sigma_{max}^{1/\rho} + \frac{t}{N-1}(\sigma_{min}^{1/\rho} - \sigma_{max}^{1/\rho}) \tag{6}$$

, with $t \in [0,1]$ and $N = 200$, the number of integration steps when using the default implementation.

In our notation we define:

$$\sigma_t = c_t, \quad \hat{\sigma}_t = \hat{t} = c_t(1 + \gamma) \tag{7}$$

Where $c_t$ and $\hat{t}$ are the terms in the original implementation. $\gamma$ is a hyperparameter set to 0.8 in the AlphaFold3 and OpenFold3 implementations, and 0.7 in the Boltz-2 implementations. $\gamma$ is set to zero for the later stages of sampling. From AlphaFold3 Algorithm 18 the update equation is:

$$x_{t+1} = x_t + \xi + \eta(\frac{c_t}{\hat{t}} - 1)(\hat{x} - x_t)(c_{t+1} - \hat{t}) \tag{8}$$

, which we also write as:

$$x_{t+1} = \tilde{x}_t + \eta(\frac{\sigma_{t+1}}{\hat{\sigma}_t} - 1)(x_t - \hat{x}_t), \tag{9}$$

with step size $\eta$, and variance terms $\sigma$ and $\hat{\sigma}$ defined by the sampling schedule.

The $\gamma$ term confounds $\hat{t}$ somewhat, however, if we impose $\gamma = 0$ then:

$$(c_{t+1} - \hat{t}) = \sigma_{t+1} - \sigma_t = \frac{d\sigma}{dt} = \dot{\sigma} \tag{10}$$

.

Further applying the relation between the score function and the denoising function $D_\theta(x_t, t) = \hat{x}$ from Equation 1:

$$x_{t+1} = x_t - \eta\dot{\sigma}_t\sigma_t s_\theta(x_t, t) + \xi \tag{11}$$

$\xi$ is the injected noise during diffusion sampling:

$$\xi = \lambda\sqrt{\hat{t}^2 - \hat{c_t}^2}\epsilon = \lambda\sqrt{2\gamma\sigma_t^2 + \gamma^2\sigma_t^2}\epsilon = \sqrt{\tilde{\sigma}_t}\epsilon, \quad \epsilon \sim \mathcal{N}(0, I_3) \tag{12}$$

, with $\lambda = 1.003$ another hyperparameter from the AlphaFold3 implementation.

At the continuous limit, the SDE of the update function is:

$$dx = -\dot{\sigma}_t\sigma_t s_\theta(x_t, t)dt + \sqrt{\tilde{\sigma}_t}dw \tag{13}$$

, where $dw$ denotes a Wiener process and we define $\tilde{\sigma}_t = \lambda^2(2\gamma\sigma_t^2 + \gamma^2\sigma_t^2)$. There is a discrepancy between the SDE used by Song et al. (2021) and the AlphaFold3 implementation, in that $\sigma$ is scaled by a constant term dependent on $\gamma$, rather than by $\dot{\sigma}$. Empirically, we find this does not obstruct the flow-sampling implementation, but the effect of this alternative noise scaling term on the marginal distributions should be investigated further.

## B   DERIVING THE PROTEIN STRUCTURE PREDICTION PF-ODE

The reverse SDE implemented in AlphaFold3 is given by Eq. 2:

$$dx = -\sigma_t\dot{\sigma}_t s_\theta(x, \sigma_t)dt + \sqrt{\sigma_t\dot{\sigma}_t}dw(t) \tag{14}$$

Comparison to the general reverse SDE (eq. 3) yields:

$$f(x, t) = 0, \quad g(t) = \sqrt{\sigma_t\dot{\sigma}_t} \tag{15}$$

This aligns with the VE forwards SDE, in which there is no drift term and noise is gradually added to the data. Inserting eq. 15 in the general form PF-ODE corresponding to the reverse SDE, given by eq. 4 yields:

$$\dot{x} = -\frac{1}{2}\sigma_t\dot{\sigma}_t\nabla_x\log p_t(x) \tag{16}$$

Finally, substituting eq. 1 yields eq. 5.

## C  FLOW-BASED STRUCTURE PREDICTION SAMPLING

---

**Algorithm 1** Flow-based structure prediction sampling

---

**Require:** Pre-trained diffusion denoising model $D_\theta$, input states $s_i^{\text{inputs}}$, trunk states $s_i^{\text{trunk}}$, trunk pair states $z_{ij}^{\text{trunk}}$, noise schedule $[c_0, c_1, \ldots, c_T]$, $\gamma_0 = 0.7$, $\gamma_{\min} = 1.0$, noise scale $\lambda = 1.003$, step scale $\eta = 1.5$

**Ensure:** Final sample $x$

1: Initialize $x \sim c_0 \cdot \mathcal{N}(0, I_3)$, where $x \in \mathbb{R}^3$
2: **for** each $c_\tau \in \{c_1, \ldots, c_T\}$ **do**
3:     $x \leftarrow R \circ x + T$
4:     $\gamma \leftarrow \begin{cases} \gamma_0, & \text{if } c_\tau > \gamma_{\min} \\ 0, & \text{otherwise} \end{cases}$
5:     $\hat{t} \leftarrow c_{\tau-1}(\gamma + 1)$
6:     $x^{\text{denoised}} \leftarrow D_\theta(x, \hat{t}, s_i^{\text{inputs}}, s_i^{\text{trunk}}, z_{ij}^{\text{trunk}})$
7:     $\delta \leftarrow (x - x^{\text{denoised}})/2\hat{t}$
8:     $\Delta t \leftarrow c_\tau - \hat{t}$
9:     $x \leftarrow x + \eta \, \Delta t \, \delta$
10: **end for**
11: **return** $x$

---

## D  ACCELERATED SAMPLING TIMES

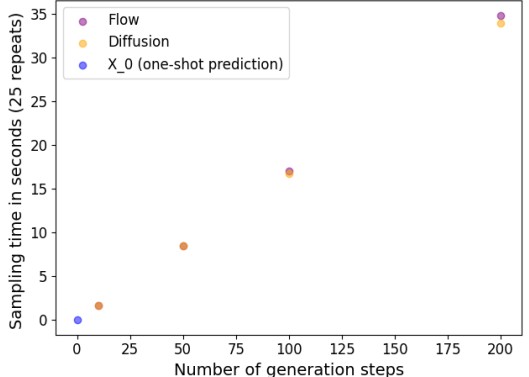

Figure 4: Structure prediction sampling times against number of steps taken throughout trajectory. Sampling times do not include the time taken to generate or embed the MSA.

# E   STRUCTURE PREDICTION EXAMPLES

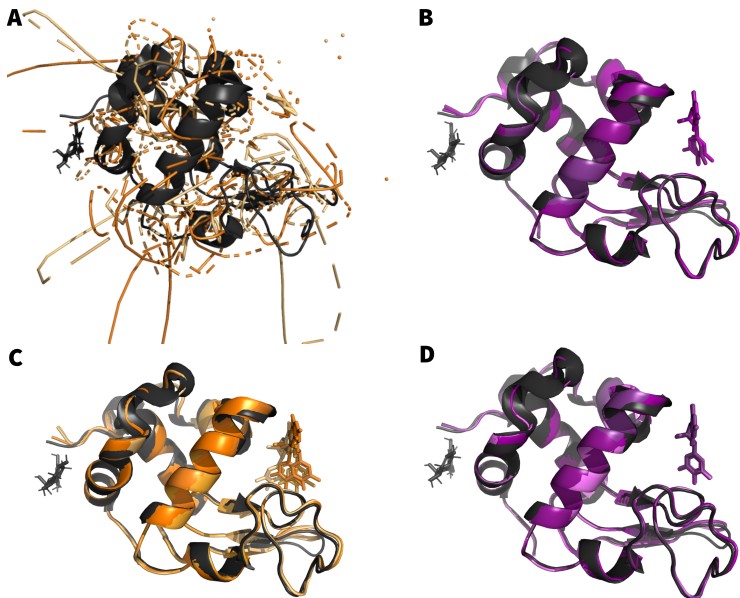

Figure 5: Reducing the number of sampling steps by an order of magnitude leads to incorrect predictions for diffusion-based sampling, whereas flow-based deterministic sampling produced accurate predictions. A) Two independent diffusion-sampled predictions made with ten diffusion steps of Boltz-2. B) Two independent flow-sampled predictions made with ten flow steps of Boltz-2. C) Four independent diffusion-sampled predictions made with 200 diffusion steps of Boltz-2. D) Four independent flow-sampled predictions made with 200 flow steps of Boltz-2. Ground truth crystal structure co-ordinates of the protein and ligand (PDB ID: 7EKA) in dark grey.

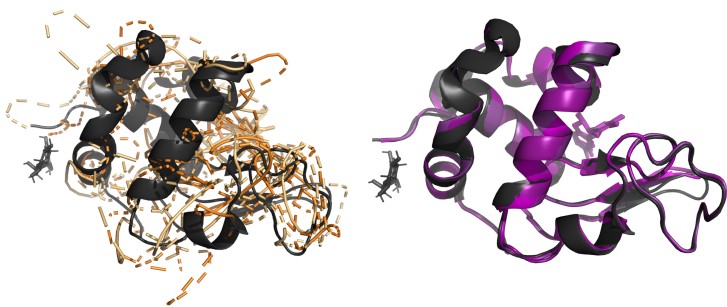

Figure 6: Diffusion- (orange) and flow-based (purple) predictions after 10 sampling steps using OpenFold3 score model (The OpenFold3 Team, 2025). Crystal structure in grey (PDB ID:7EKA)

# F   FOLDBENCH BENCHMARK EXTENDED RESULTS

Extended results of the FoldBench benchmark.

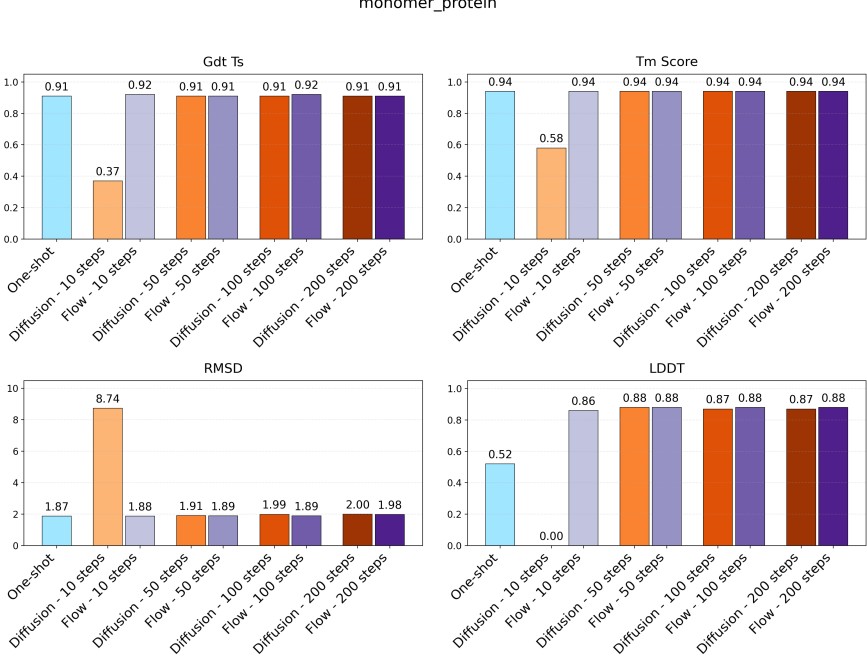

Figure 7: FoldBench results on monomer protein set.

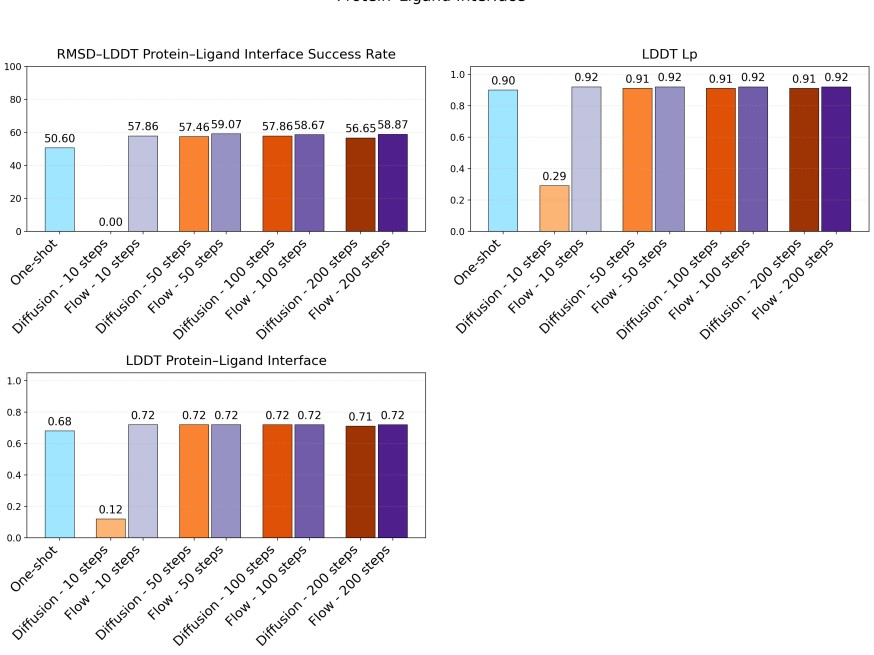

Figure 8: FoldBench results on protein-ligand set.

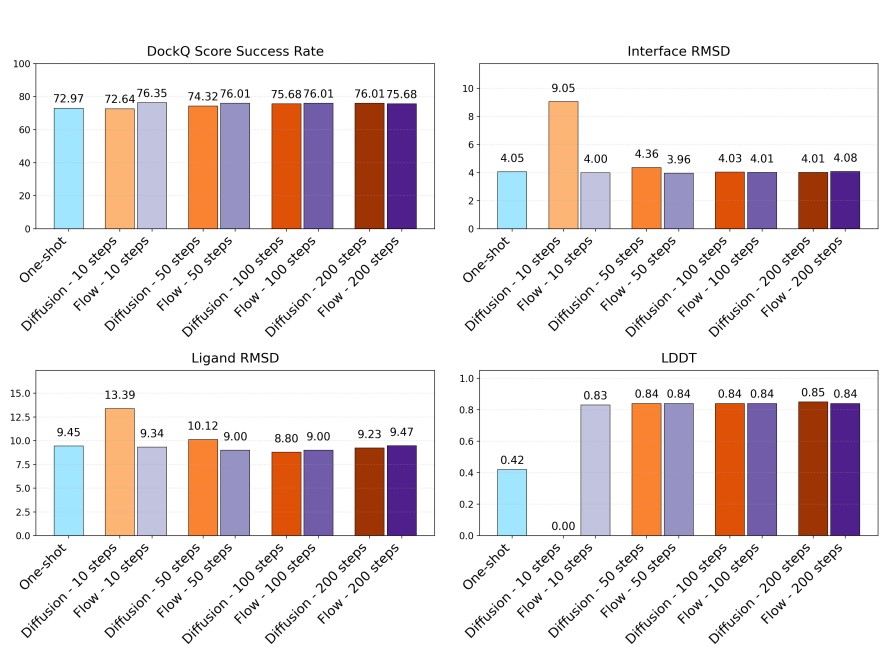

Figure 9: FoldBench results on antibody:antigen set.

Figure 10: FoldBench results on protein-DNA co-folding set.

## G  FOLDBENCH TO BOLTZ-2 OVERLAP SEVERITY

The FoldBench test set uses a cut-off on 2023-01-13, whereas Boltz-2 was trained on data released before 2023-06-01. While this does not effect comparisons between diffusion and flow-matching sampling performance, the observed over-conditioning effects are likely stronger for any overlapping examples. We report the proportion of test-set examples released after the Boltz-2 cut-off in Table 2.

| Test set type | Fraction of FoldBench test set released after 2023-06-01 | Percentage of FoldBench test set released after 2023-06-01 |
|---|---|---|
| Ab:Ag complexes | 134 / 172 | 77.91% |
| Monomer structures | 260 / 334 | 77.84% |
| Protein-ligand complexes | 445 / 558 | 79.75% |

Table 2: Proportion of FoldBench test examples released after the Boltz-2 training cut-off. The majority of test examples are uneffected.

## H  PREDICTION DIVERSITY

Extended version of Table 1.

| | Flow | | | | | | | | Diffusion | | | | | | | |
|---|---|---|---|---|---|---|---|---|---|---|---|---|---|---|---|---|
| Nr of integration steps | 10 | | 50 | | 100 | | 200 | | 10 | | 50 | | 100 | | 200 | |
| Redundancy threshold (Å) | 1 | 5 | 1 | 5 | 1 | 5 | 1 | 5 | 1 | 5 | 1 | 5 | 1 | 5 | 1 | 5 |
| Mean average | 1.0 | 1.0 | 1.0 | 1.0 | 1.0 | 1.0 | 1.0 | 1.0 | NA | NA | 2.3 | 1.0 | 2.3 | 1.0 | 2.3 | 1.0 |
| Mode average | 1 | 1 | 1 | 1 | 1 | 1 | 1 | 1 | NA | NA | 1 | 1 | 1 | 1 | 1 | 1 |
| 7eka | 1 | 1 | 1 | 1 | 1 | 1 | 1 | 1 | NA | NA | 5 | 1 | 7 | 1 | 7 | 1 |
| 7zlo | 1 | 1 | 1 | 1 | 1 | 1 | 1 | 1 | NA | NA | 5 | 1 | 4 | 1 | 4 | 1 |
| 8atl | 1 | 1 | 1 | 1 | 1 | 1 | 1 | 1 | NA | NA | 1 | 1 | 1 | 1 | 1 | 1 |
| 8jjt | 1 | 1 | 1 | 1 | 1 | 1 | 1 | 1 | NA | NA | 1 | 1 | 1 | 1 | 1 | 1 |
| 8jny | 1 | 1 | 1 | 1 | 1 | 1 | 1 | 1 | NA | NA | 1 | 1 | 1 | 1 | 1 | 1 |
| 8tfq | 1 | 1 | 1 | 1 | 1 | 1 | 1 | 1 | NA | NA | 2 | 1 | 1 | 1 | 1 | 1 |
| 8u8j | 1 | 1 | 1 | 1 | 1 | 1 | 1 | 1 | NA | NA | 1 | 1 | 1 | 1 | 1 | 1 |

Table 3: Expanded version of Table 1. Number of non-redundant ligand poses explored across 25 independent samples, when using different redundancy thresholds. Diffusion-based sampling explores more local ligand conformations than flow-based sampling, however neither method explores multiple binding pockets.

