# OpenReview forum: "Converting diffusions to flows accelerates sampling and suggests over-conditioning of co-folding models on sequence"
_ICLR.cc/2026/Workshop/LMRL — ICLR 2026 Workshop LMRL Poster_

### Official Review · Reviewer_Xf4i · 2026-02-24
**The paper takes diffusion-SDE sampling used by AlphaFold3-style co-folding predictors and converts it to a probability-flow ODE sampler, using the same pretrained score network. This concept / idea is well known and has been done before, but the results are of interest as they demonstrate faster sampling for co-folding models is possible.**

**Rating:** 5
**Confidence:** 4

**Review:**

Pros:

The paper shows that these co-folding generative models do not use the continuous noise space to explore diverse candidate structures, and act like point-mass distributions heavily over-conditioned on sequence and ligand embeddings. This fact has been backed up by other works such as [6] and [7]. It is useful to see that these results have been confirmed by other works and confirm these other works. This demonstration is not novel, however.

They provide a more efficient sampling method for co-folding models, which is useful.

Cons:

Algorithmically, there is no novelty here; only in application to co-folding models. The conversion is well established in score-SDE theory [1].

The paper compares PF-ODE Euler-style sampling at 10 steps to the original diffusion discretization at 10 steps, which predictably degrades. There is an abundance of literature for fast diffusion sampling, such as [3], [4], [5]. Without this comparison it is hard to say whether or not the acceleration is from the ODE path, better numerical integration, etc. Both [4] and [5] can reduce the number of steps taken to ~10-20 and get high quality results.

I'm not sure if I agree that not including extra scaling / hyperparamters does not affect the flow implementation, and additionally, the effective continuous-time SDE implied by the discrete update is not the canonical VE SDE used in [1]. In AF3, it seems as though the marginal distribution is emergent from the learned discrete dynamics, which means we we not understand them analytically, and thus there isn't a theoretically correct PF-ODE (well at least not one that can be shown without a proof; it is possible for this to occur). What the authors have defined is a loose approximation.


Overall:

In terms of clarity, I think the paper is readable and has a coherent narrative. The trajectory + "one-shot" predictions is well communicated. I am glad that the authors do not claim much novelty, as most of this work has been done before, but I do appreciate the application to co-folding models.
In terms of quality, I think the evaluation on FoldBench is fair, and they demonstrate that their sampling algorithm maintains parity with the diffusion baseline. Other qualms were addressed in the cons section.
Originality is very low. The equations for migrating from diffusion to PF-ODE are standard, the efficient sampling idea has already been looked at, and the results that show these models lack diversity has been discussed in other papers.
The results do merit some points in terms of significance, because it is useful to know that we can efficiently speed up sampling performance and get good one-shot predictions using a different sampler in the case of protein design.

I think my qualms outweigh the pros, but as it is a workshop, I won't be upset to see this accepted. I do hope significant clarifications and additions to this work are added in the future. I would recommend a weak reject.


[1]: Song, Yang, et al. "Score-Based Generative Modeling through Stochastic Differential Equations." International Conference on Learning Representations.

[2]: Karras, Tero, et al. "Elucidating the design space of diffusion-based generative models." Advances in neural information processing systems 35 (2022): 26565-26577.

[3]: Lu, Cheng, et al. "Dpm-solver: A fast ode solver for diffusion probabilistic model sampling in around 10 steps." Advances in neural information processing systems 35 (2022): 5775-5787.

[4]: Song, Jiaming, Chenlin Meng, and Stefano Ermon. "Denoising Diffusion Implicit Models." International Conference on Learning Representations.

[5]: Lu, Cheng, et al. "Dpm-solver++: Fast solver for guided sampling of diffusion probabilistic models." Machine Intelligence Research 22.4 (2025): 730-751.

[6]: Rosenberg, A. A., Vedula, S., Bronstein, A. M., and
Marx, A. Seeing double: Molecular dynamics
simulations reveal the stability of certain alter-
nate protein conformations in crystal structures.
bioRxiv, 2024b. doi: 10.1101/2024.08.31.610605.
URL https://www.biorxiv.org/content/
early/2024/08/31/2024.08.31.610605.

[7]: Maddipatla, Sai Advaith, et al. "Inverse problems with experiment-guided AlphaFold." Forty-second International Conference on Machine Learning.

---

### Official Review · Reviewer_fRxR · 2026-02-26
**Theory is incomplete, admirable experimental effort**

**Rating:** 5
**Confidence:** 2

**Review:**

Note: The reviewer is theory inclined and has weak opinions on the value of the experiments, the area chair should override my decision if the experiments are of value.

The theoretical sections of this paper is incomplete, the drift and diffusion terms defined for the reverse process nor is the score function defined. The text says the score is the gradient of the logarithmic of a density function p_t, but how is that defined? With sufficient care, these statements can be made precise, but is not currently present.

Meaningful metrics are reported, and the claim about speed seems justified by experimental evidence.

---

### Official Review · Reviewer_frpd · 2026-02-26

**Rating:** 8
**Confidence:** 5

**Review:**

The authors utilize a deterministic flow process to accelerate the sampling of folding models, namely OpenFold and Boltz2. They find in doing so the number of steps necessary to achieve SOTA performance decreases from 200 to 10 and the majority of folding is done in the middle of the trajectory.

The empirical results are quite compelling. Reducing the number of sampling steps from 200 to 10 while maintaining strong performance is a substantial practical improvement and represents a meaningful speedup over the current standard. The analysis of the folding trajectory is interesting, particularly the observation that much of the apparent structural formation occurs by the middle of the trajectory. This is then supported by the one-shot results which question if folding model utilize the noisy coordinates at all, instead relying on the input sequence embeddings. If this result is robust and reproducible, it could be a useful insight for the community and may connect to broader ongoing discussions about how these models perform inference over time.

My main reservation is reproducibility. The claims are strong, but I did not see accompanying code, which makes it harder to assess how easily others will be able to verify and build on these results. Still, given the practical significance of the reported speedup and the potential broader relevance of the trajectory analysis, I believe this work makes a meaningful workshop contribution.

---

### Meta-Review · Area_Chair_2f2r · 2026-02-27

**Recommendation:** Accept (Poster)
**Confidence:** 3

**Metareview:**

Accept.

---

### Decision · Program_Chairs · 2026-03-02

**Decision:**

Accept (Poster)

**Comment:**

Please see the meta-review.